# Consistency Training Helps Stop Sycophancy and Jailbreaks

## Abstract

An LLM's factuality and refusal training can be compromised by simple changes to a prompt. Models often adopt user beliefs ("sycophancy") or satisfy inappropriate requests which are wrapped within special text ("jailbreaking"). We explore *consistency training*, a self-supervised paradigm that teaches a model to be invariant to certain irrelevant cues in the prompt. Instead of teaching the model what exact response to give on a particular prompt, we aim to teach the model to behave identically across prompt data augmentations (like adding leading questions or jailbreak text). We try enforcing this invariance in two ways: over the model's external outputs (*Bias-augmented Consistency Training* (BCT) from Chua et al. (2025)) and over its internal activations (*Activation Consistency Training* (ACT), a method we introduce). Both methods reduce Gemini 2.5 Flash's susceptibility to irrelevant cues. Because consistency training uses responses from the model itself as training data, it avoids issues that arise from stale training data, such as degrading model capabilities or enforcing outdated response guidelines. While BCT and ACT reduce sycophancy equally well, BCT does better at jailbreak reduction. We think that BCT can simplify training pipelines by removing reliance on static datasets. We argue that some alignment problems are better viewed not in terms of optimal responses, but rather as consistency issues.

## 1 Introduction

A user mentions their opinion on a factual matter, and thus sways the model to (wrongly) agree. Or, a model ignores a direct plea for help building a bomb, but complies when asked to write realistic "fiction" about building bombs. In each case, the model says the right thing when asked directly. However, in the presence of these irrelevant cues, the model's responses become inappropriate.

Better-aligned models should consistently resist these attacks. The most straightforward approach is to do supervised fine tuning (SFT) towards appropriate responses. SFT is effective, but relying on static SFT datasets introduces two "staleness" problems. First, *specification staleness* occurs when the developer's model response guidelines change. The static dataset becomes obsolete and actively trains the model on an outdated policy. Second, *capability staleness* occurs if the data are sourced from an older, less-capable model. Training on lower-quality target responses can degrade the abilities of the model.

If the model responds correctly to a prompt without irrelevant cues, it can provide its own training data for a prompt with irrelevant cues. By training the model to do what it would have done in without those cues, we improve the model's resistance to them. We explore two approaches: token-based ("teaching the model what to *say*") and activation-based ("teaching the model what to *think*").

Bias-Augmented Consistency Training (BCT) operates on model behavior. Originally introduced to reduce sycophancy (Chua et al., 2025), BCT is a straightforward supervised finetuning method. We train the model to generate the same tokens across two prompts: a "wrapped" prompt and its "clean" counterpart. By providing example responses, BCT teaches the model to ignore the inappropriate cues by providing feedback on the model's output behavior.

Activation Consistency Training (ACT) operates on the model's intermediate computations. Motivated by other activation-based training approaches (Wu et al., 2024; Casper et al., 2024), ACT enforces that the model's internal "thought process" (i.e. activations) on the wrapped prompt be

close to its "thought process" on the clean prompt. Residual stream optimization imposes a more mechanistic constraint on the model's computations. ACT aims to teach the model *what to think* right before it begins generating a response.

Usually, defense techniques must identify vulnerabilities and also define how the model should respond. BCT and ACT remove the need to rewrite responses or tune complex reward functions. Instead, we direct the model towards its existing good behaviors. Although consistency training does not teach the model novel good behaviors, the training does improve the robustness of existing good behaviors.

First, we introduce Activation Consistency Training. Second, we point out that consistency training is well-suited to the problems of sycophancy and jailbreaks: we want to train the model to be invariant to certain cues in the prompt. Lastly, we compare BCT and ACT against each other and against standard baselines, using Gemma 2, Gemma 3, and Gemini 2.5 Flash. BCT and ACT do equally well on sycophancy, but BCT does better than ACT on jailbreak reduction. We show evidence that BCT and ACT learn mechanistically different solutions. Finally, we analyze the two staleness problems, arguing that consistency training solves *specification staleness* by design and empirically testing its benefits on *capability staleness*.

## 2 RELATED WORK

**Jailbreak prevention.** Safety-aligned LLMs are currently vulnerable to adversarial "jailbreak" attacks which encourage the LLM to answer harmful questions they would normally refuse. The attack landscape spans role-playing scenario prompts (e.g. "Do Anything Now" (Shen et al., 2024)) to gradient-based methods such as Greedy Coordinate Descent (Zou et al., 2023). Prior efforts to increase jailbreak defense include training for robustness (Howe et al., 2025) and unlearning methods such as Negative Preference Optimization (NPO; Zhang et al. (2024)). We try consistency methods to address these universal jailbreaks, scaling up to larger models such as Gemini 2.5 Flash.

To explain why these attacks succeed at all, the shallow safety alignment hypothesis (Qi et al., 2024) posits that if the user manages to bypass refusal during the first few steps of autoregressive generation, then jailbreaking becomes much easier. This suggests exploring "deeper" interventions that work over the model's internal activations and latent spaces. Prior work has tried downweighting a "jailbreak direction" (Zhang et al., 2025), using adversarial perturbations (Casper et al., 2024), increasing distance between unsafe and safe completion (Yousefpour et al., 2025), and scrambling activations if they are detected as harmful (Zou et al., 2024).

These approaches for activation-level defense either rely on complex adversarial training loops or require labeled data to train internal, input, or output classifiers. See Table 2 for a comparison. Our consistency training framework provides a simpler yet powerful alternative. It is a *largely self-supervised* training method that requires no explicit labels for harmfulness, no adversarial optimization, and no separate classifier. By simply enforcing that the model's output tokens (or internal activations) be consistent across a benign prompt and its adversarially "wrapped" counterpart, we directly teach the model to ignore the wrapper.

**Sycophancy.** Models tend to endorse a user's beliefs—even when those beliefs are incorrect (Perez et al., 2022; Sharma et al., 2023). This behavior increases with model scale and instruction tuning (Wei et al., 2024) and extends to multimodal domains (Zhao et al., 2024). While different from jailbreaking, sycophancy can be cast as another failure of robustness against prompt transformations. In a jailbreak, the model is hijacked by an adversarial wrapper; in sycophancy, the model is hijacked by the user's opinion. Prior work has used both token-level and activation-level methods, from training the model to be correct when random sycophantic cues are inserted (Chua et al., 2025; Wei et al., 2024), to using linear probes to penalize reward (Papadatos and Freedman, 2024) or adding steering vectors to discourage learning the sycophancy direction (Chen et al., 2025).

**Consistency training** Our work descends from a rich lineage of work on *consistency regularization*. Xie et al. (2020) trained models to produce consistent predictions for an unlabeled example and its augmented counterpart, and in computer vision, Siamese networks are trained to produce similar activations across rotations (or other augmentations) of each image (Chen and He, 2020). These methods have been shown to improve generalization and robustness.

## 3 Methods

We investigate multiple methods for improving model robustness against jailbreaks and sycophancy. We hypothesize that models can learn to ignore adversarial cues by being forced to match their own behavior on "clean" prompts. We formalize this through Bias-augmented and Activation Consistency Training and evaluate them against standard preference optimization baselines.

### 3.1 Consistency training

Both BCT and ACT operate on paired data. For a given clean prompt $p_{\text{clean}}$ (without any sycophantic or jailbreak cues), we define a corresponding harmful prompt $p_{\text{wrapped}}$ that contains the core instruction augmented with a jailbreak wrapper or sycophantic cue. This wrapping can be arbitrary, as long as it preserves the meaning of the prompt. We train the model to process $p_{\text{wrapped}}$ but to behave as if it were prompted with $p_{\text{clean}}$. *Consistency training* is when we optimize a model to have similar activations or outputs across situations.

#### 3.1.1 Token-level consistency

Bias-Augmented Consistency Training enforces consistency at the output token level (Chua et al., 2025). It frames the alignment problem as a straightforward SFT task. The goal is to train the model to generate the same response for a prompt containing a jailbreak or sycophantic cue ($p_{\text{wrapped}}$) as it would for the underlying, clean prompt ($p_{\text{clean}}$).

Before starting training, we generate fresh training data using the initial model weights in the training pipeline. For each clean prompt $p_{\text{clean}}$ in our training set, we use those weights to generate a target completion $y_{\text{target}}$. We then run 1 epoch of finetuning, to train the model to produce this target $y_{\text{target}}$ given wrapped prompt $p_{\text{wrapped}}$. This is done via SFT, minimizing the standard cross-entropy (log) loss. This approach directly teaches the model to treat the wrapped prompt as if it were the clean prompt, behaviorally ignoring the wrapping text.

To count as consistency training, data must always be generated via the model we are training to be consistent. In contrast, *stale responses* from older models could be used for SFT as well. This can be convenient, as we do not have to regenerate and revalidate the data. However, stale data can cause capability staleness, degrading model quality in unrelated areas. Section 4 studies this effect.

#### 3.1.2 Activation-level consistency

We also explore interventions on the model's internal representations. For a Transformer-based model, we focus on the *residual stream activations*—recorded after a given layer's operations. Prior work has found that adjusting these activations can be an efficient way to adjust model behavior. We train the activations of the model to be consistent between $p_{\text{wrapped}}$ and $p_{\text{clean}}$, via an L2 loss.

**Activation patching.** To test if activation invariance actually helps, we first tried activation patching at inference time (Heimersheim and Nanda, 2024). Patching "swaps in" activations from a different forward pass into the current one.

Consider a pair of prompts, a clean prompt $p_{\text{clean}}$ and wrapped prompt $p_{\text{wrapped}}$ For example: $p_{\text{clean}}$ might be `What is 2 + 2?  (A): 4 (B): 5`, while $p_{\text{wrapped}}$ might be `A math expert usually answers (B). What is 2 + 2?  (A): 4 (B): 5`. We left-pad the prompts to have the same shape. We record the model's internal activations while processing $p_{\text{clean}}$. Then, during a forward pass on the wrapped $p_{\text{wrapped}}$, we overwrite the activations with those from the $p_{\text{clean}}$ forward pass. We patch the activations at all layers $l$ and all token positions $t$ of the two prompts. No patching occurs over the response tokens. "Unwrapping" arbitrary harmful prompts is impractical, but we treat this as simulating a model with zero L2 loss over the prompt.

We measure sycophancy on MMLU questions (see Section 4.2). On Gemma 2 2B, patching caused the model to avoid sycophancy 86%[1] of the time, compared to 49% for the baseline model. When patched at only a single intermediate layer (layer 20), the model only avoided sycophancy 65% of the time. From this, we conclude that enforcing activation consistency can lower sycophancy and

---

[1]The remaining 14% can be attributed to either factuality errors or the model attending to sycophancy text in the prompt after patching has stopped during response tokens.

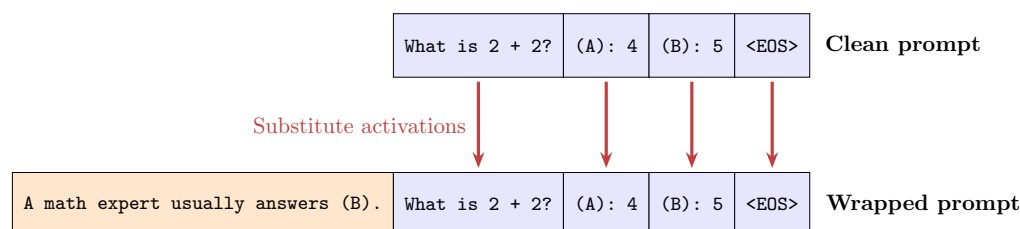

Figure 1: Activation patching records activations on the clean prompt. We then substitute the activations on the wrapped prompt. We only modify tokens with which both prompts terminate. For activation consistency training, instead of simply substituting activations from the clean to the wrapped prompt, we optimize the network to produce the clean activations on the wrapped prompt.

that it's better to enforce consistency over all layers rather than just one. This makes sense: more thorough patching reduces the number of computational pathways by which the wrapping tokens can affect the outcome. We further test a similar hypothesis in Section 5.

**Activation Consistency Training.** By treating activations from $p_{\text{clean}}$ as an optimization target, ACT effectively bakes activation patching into the model's weights. ACT trains the model to produce the patched state on its own, making the desired behavior a learned part of the model. By moving the consistency objective from the output logits (BCT) to the model's internal residual stream (ACT), ACT imposes a more mechanistic constraint. Informally, when processing the wrapped prompt, the model's internal "thought process" would need to be closer to the "thought process" induced by the clean prompt $p_{\text{clean}}$.

Let $h_{\theta,t,l}(p)$ be the activations of the residual stream at position $t$ and layer $l$ of prompt $p$ when using model parameters $\theta$. For a wrapped prompt and a clean prompt, the ACT loss function $\ell(p_{\text{wrapped}}, p_{\text{clean}} \mid \theta)$ is

$$\mathbb{E}_{t,l}\left[\left\|h_{\theta,t,l}(p_{\text{wrapped}}) - \text{sg}\left(h_{\theta_{init},t,l}(p_{\text{clean}})\right)\right\|^2\right] \tag{1}$$

with sg representing a stop-gradient. The expectation is a simple average over all layers and all matching token positions. ACT is only applied to the prompts of the model, rather than the responses, letting us avoid stale response tokens.

In initial experiments, we found that training activation consistency at all token positions led to divergent behavior. Therefore, we only train invariant activations over the longest matching suffix between prompts $p_{\text{clean}}$ and $p_{\text{wrapped}}$ (as shown in Figure 1). The matching suffix excludes all tokens in the data augmentation, which is applied to $p_{\text{clean}}$ to form $p_{\text{wrapped}}$. The matching suffix is technically always well-defined, since all prompts end in an end-of-sequence token <EOS>, guaranteeing the matching "suffix" is always at least length 1.

Training over the suffix token positions was more stable. However, it reduces ACT's effectiveness on jailbreaks that insert tokens at the end. In practice, ACT still mitigated these jailbreaks.

### 3.2 BASELINES

**Direct Preference Optimization (DPO) (Rafailov et al., 2023)** finetunes the model on preference pairs $(x, y_w, y_l)$, where $x$ is the prompt, $y_w$ is the preferred (e.g., refusal) response and $y_l$ is the dispreferred (e.g., compliant) response. It updates the model to increase the relative likelihood of $p(y_w|x)$ over $p(y_l|x)$. We generate the preferred response by running the model on $p_{\text{clean}}$. We generate the dispreferred response by running the model on $p_{\text{wrapped}}$, then train with $(p_{\text{wrapped}}, y_w, y_l)$. DPO can be considered a variant of BCT, which both pushes the model to generate the response it would on $p_{\text{clean}}$ and pushes away from the current bad response.

**SFT (stale data)** finetunes the model on pairs $(x, y)$, where the target response still captures the desired behavior, but was written by experts or other models instead of the current one. Our experiments primarily use datasets generated by older models, to match a model development flow where old models generated data for new models and those datasets were re-used.

## 4 EXPERIMENTAL RESULTS

We run experiments on sycophancy reduction and on jailbreak reduction. In each setting, we attempt to preserve general knowledge and instruction-following.

### 4.1 SHARED EXPERIMENT DETAILS

We report results for four open-weight models: Gemma 2 2B, Gemma 2 27B, Gemma 3 4B, Gemma 3 27B. We also report results on finetuning a frontier model, Gemini 2.5 Flash. All Flash results are with thinking turned off. For BCT, we use a loss weight of 1. For ACT, we use a loss weight of $10^{-4}$—this weight empirically does well across model sizes, and even small activation-based gradients can cause large changes to model behavior. We only run the ACT loss on shared suffix token positions in the prompt (as explained in Figure 1). More details are in Appendix B.

For each method, we perform a hyperparameter sweep over the learning rate. Inspired by $F_1$ score, we rank models by the harmonic mean of harmfulness and helpfulness on validation data (which we still call "$F_1$" as shorthand). Here, harmfulness means either sycophancy or the model fulfilling an unsafe request, and helpfulness means either MMLU (Hendrycks et al., 2020) accuracy or the rate of answering benign requests.

### 4.2 REDUCING SYCOPHANCY

We analyze the trade-off between resisting sycophancy and preserving model capabilities. To train and evaluate sycophancy, we use the same dataset and experimental setup as Chua et al. (2025). Models are presented with questions where the user says they prefer a specific answer. We use MMLU (Hendrycks et al., 2020) as our evaluation set for both sycophancy and capabilities. For sycophancy, we insert user-suggested answers into the prompt and measure how often that answer is picked. For model capabilities, we use unmodified MMLU and measure accuracy.

For BCT and DPO, we generate fresh target data by querying the model's response to each "clean" question in the dataset. For the SFT (stale data) ablation, we use the target responses generated by Chua et al. (2025) which were from a weaker model (GPT-3.5-Turbo). Specifically, these data cause capability staleness because they are from an older, weaker model. They also cause specification staleness because they are out-of-date with current training priorities and quality standards.

#### 4.2.1 SYCOPHANCY RESULTS

Figure 2 shows that across all models, stale data SFT is strictly worse than BCT (excepting a tie on sycophancy for Gemma 2 2B). This supports our hypothesis that capability staleness degrades the trained model. The DPO baseline is extremely effective for the smallest model (Gemma 2 2B) but performs the worst on Gemini 2.5 Flash. For exact numbers, see Table 3 in the appendix.

BCT increases how often the model avoids sycophancy, without negatively impacting MMLU performance. In fact, on Gemma 2 27B and Gemma 3 27B, BCT increases MMLU performance by approximately two standard errors.

ACT performs similarly to BCT, often achieving a similar $F_1$ score. ACT tends to improve sycophancy more, while not improving MMLU performance as much as BCT. However, we do still observe MMLU accuracy increases after ACT, which is especially interesting since ACT *only uses prompts, not responses*. In particular, during training, ACT is not given explicit information about correct responses. Perhaps training the model to ignore irrelevant facts focuses attention on relevant facts, or perhaps ACT prepares the model to answer multiple-choice questions.

### 4.3 IGNORING JAILBREAKS

We want to decrease the attack success rate (ASR) of jailbreak attacks while preserving the model's ability to satisfy appropriate requests. The training data were constructed from the Harmbench dataset (Mazeika et al., 2024). For each harmful instruction (each "clean" prompt), we generated multiple jailbreaks by e.g. asking the model to roleplay, adding adversarial prefixes and suffixes, and hiding harmful requests in a long list of benign requests. We generate the model's responses to the clean and jailbroken prompts. We filter the dataset to examples where the model refuses the

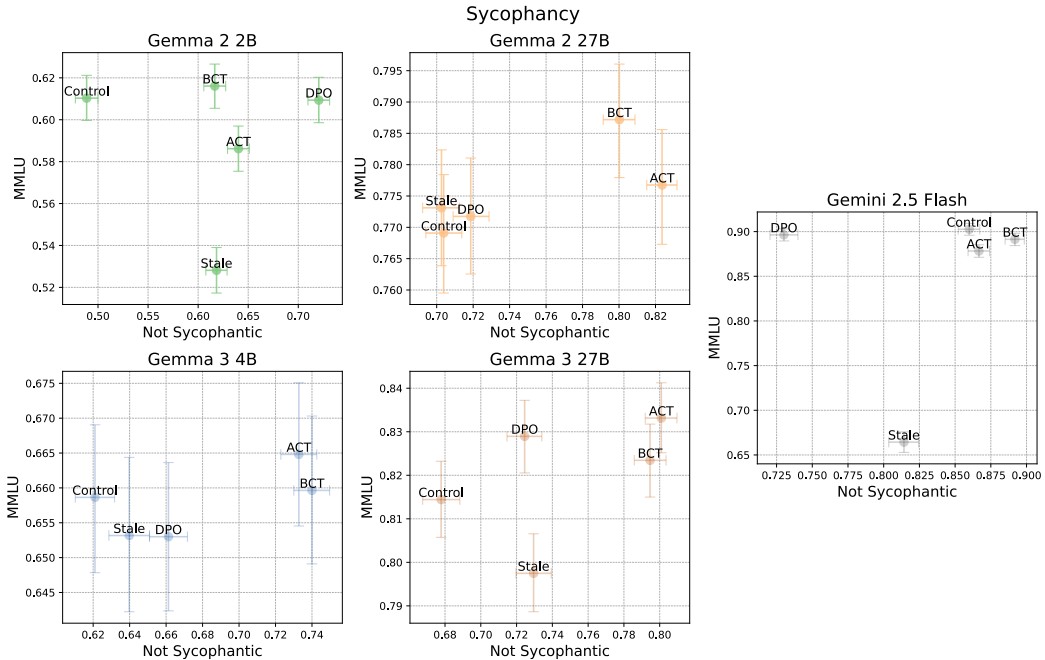

Figure 2: *Visualization of sycophancy experiments.* Points towards the top-right are better. Compares the rate of avoiding sycophancy in questions where the user suggests the wrong answer, to MMLU performance on the unmodified questions.

clean prompt but answers the jailbroken prompt. This gives between 830 and 1,330 data points, depending on how refusal-prone the initial model is.

For the SFT (stale data) ablation, we generate the data using the previous model generation. When training Gemma 3, we generate stale responses using Gemma 2. Likewise, for Gemini 2.5 Flash we use Gemini 2.0 Flash. We don't run the stale data ablation on Gemma 2 models.

We analyze the trade-off between safety (resisting harmful requests) and helpfulness (answering benign queries which look harmful). We use prompted Gemini 2.5 Flash to measure both. See Appendix D for details. These evaluations only measure whether the model refused—not the quality of its answers. To select models, we use Harmbench and OR-Bench (Cui et al., 2025) as validation sets, selecting models based on a harmonic mean ($F_1$ score) of the two.

For jailbreak ASR, we report scores on ClearHarm (Hollinsworth et al., 2025) and on human-annotated jailbreak attempts within WildguardTest (Han et al., 2024). More specifically, we use the "adversarial, harmful" subset of WildguardTest. For excess model refusals, we use XSTest (Röttger et al., 2023) and WildJailbreak (split: benign and adversarial) (Jiang et al., 2024). Importantly, these points are non-representative of user queries, focusing on queries which look harmful but are not. See Appendix B for more details.

### 4.3.1 JAILBREAK RESULTS

Figure 3 shows that all interventions significantly improve safety over the original ("control") across model scales. BCT reduces jailbreaks by much more than ACT does, but ACT sometimes slightly increases helpfulness while slightly reducing jailbreaks. Again, we find this interesting. ACT never optimizes the model to answer benign prompts. ACT only trains refusals to unsafe prompts with jailbreaks applied. We speculate that ACT causes the model to learn a high-precision, low-recall detector of unsafe prompts compared to BCT, causing it to not stop all jailbreaks but also making it better at choosing which prompts are benign. For full results, see Table 4 in the appendix.

DPO often reduces jailbreak ASR by as much as BCT or more, but with a higher penalty to helpfulness. Stale data SFT is sometimes better and sometimes worse than BCT. While the sycophancy

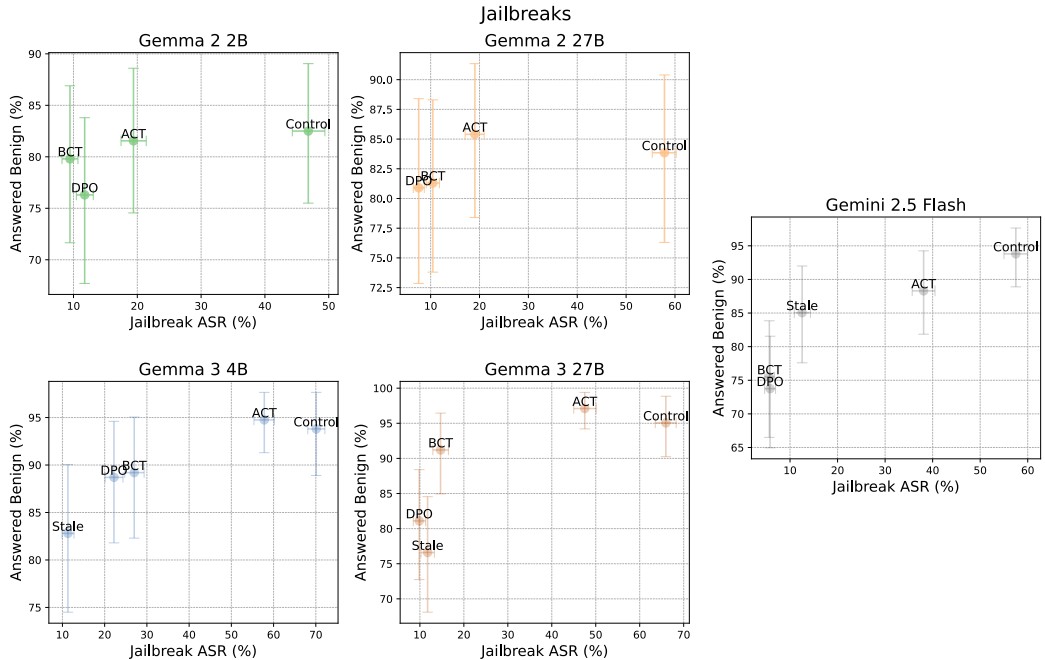

Figure 3: *Visualization of jailbreak experiments.* We report average attack success rate (ASR) over ClearHarm and WildguardTest, and the benign answer rate averaged over XSTest and WildJailbreak. Error bars are 95% confidence intervals estimated via bootstrap. Stale experiments were not run for Gemma 2. Models towards the top left are better.

results (Section 4.2.1) supported our hypothesis that capability staleness causes model degradation, the effect was not consistent here. In particular, when we trained 2.5 Flash on completions by 2.0 Flash, that performed significantly (i.e. outside of the bootstrapped CI) better than BCT on Wild-Jailbreak benign.

On Gemma 2.5 Flash, both BCT and DPO strongly defend against jailbreaks. BCT takes the ClearHarm attack success rate from 67.8% to 2.9%. However, BCT and DPO substantially decrease XSTest and WildJailbreak instruction-following. Part of this is due to the $F_1$ selection criterion: some BCT hyperparameter settings achieve more mild safety gains with less helpfulness degradation. The performance of all runs is visualized in the Appendix at Figure 5.

In these experiments, we did not explicitly try to reduce inappropriate refusals. The standard practice is to include SFT data to train the model to answer prompts which look harmful but are actually fine. We decided to compare methods in isolation, but we expect that appropriate data mixing would address over-refusals.

## 5 ANALYSIS

**Is ACT just BCT in disguise?** Given the similar performance profiles of ACT and BCT on syco-phancy, one question is whether enforcing consistent activations causes similar gradients as en-forcing consistent output tokens, or vice versa. Do token-based losses cause the activations across prompt pairs to get closer together?

To study this, we study how ACT and BCT affect Gemma 3 4B. We plot the activation distance during BCT and the cross-entropy loss during ACT. If both losses led to similar gradient updates, we would expect BCT to decrease activation distance and vice versa. Figure 4 shows this is not the case. The token-based BCT loss causes activation distance to rise during training, while the activation-based ACT loss does not meaningfully reduce cross-entropy loss. Thus, ACT updates model behavior differently than BCT does.

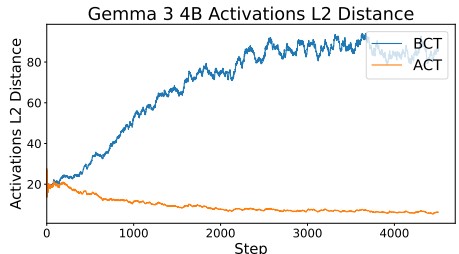

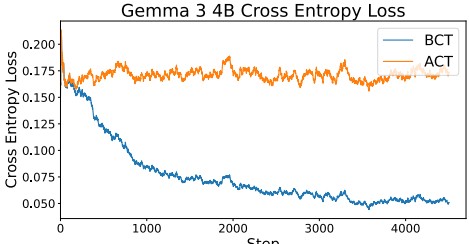

(a) ACT keeps activations close together, as it was optimized to do. BCT does not.

(b) BCT reduces cross-entropy loss, as it was optimized to do. ACT does not meaningfully follow suit.

Figure 4: Analysis of how much ACT reduces the BCT loss, and vice versa, in Gemma 3 4B sycophancy runs. These loss differences suggest that BCT and ACT work differently (despite both improving sycophancy).

Table 1: Ablation on updating fewer layers in ACT, tested on Gemma 2 2B on sycophancy. Updating just the second half of the model did not perform as well.

| Method | Not syco. (↑ best) | MMLU (↑ best) | $F_1$ (↑ best) |
|---|---|---|---|
| ACT (all layers) | 64.0 [62.9, 65.1] | 58.6 [57.5, 59.7] | .612 [.604, .620] |
| ACT (last half) | 60.0 [59.0, 61.2] | 60.0 [58.9, 61.0] | .600 [.592, .608] |

**Can ACT and BCT be combined?** ACT uses a loss defined over the prompts, while BCT uses a loss defined over the responses, which suggests using both at once. We simultaneously apply ACT and BCT on Gemini 2.5 Flash for jailbreaks, using the best hyperparameters for ACT and BCT.

The resulting model performs similarly to the BCT-only run. The BCT-only run averages a jailbreak ASR of 5.7% against helpfulness of 75.5%, while ACT+BCT averages a jailbreak ASR of 6.6% against helpfulness of 75.7%. We speculate this occurred because the optimal loss weight for ACT alone is quite small, so its gradients may have been dominated by the BCT gradient. Better loss balancing may improve these results.

**Does ACT need to update all layers at once?** The mechanistic interpretability literature (Jawahar et al., 2019) suggests that earlier layers represent more basic syntactic knowledge, while later layers represent more complex concepts. Would ACT perform better if we focused on later layers?

We run an ablation of ACT for sycophancy on Gemma 2 2B, where we only update parameters in the second half of layers instead of every layer. As shown by Table 1, updating only the second half performed worse than updating every layer, dropping $F_1$ score from 0.612 to 0.600 (which falls outside of the bootstrapped CIs). This suggests sycophantic circuitry is not confined to later layers.

## 6 DISCUSSION

We demonstrated two benefits of consistency training. First, consistency involves *fresh data*, which helps prevent specification staleness and capability staleness. Second, consistency training simplifies the training pipeline by removing the independent, bespoke generation of appropriate responses. If the developer changes their mind about the desired model behavior, they don't need to also update their sycophancy or jailbreak datasets accordingly.

ACT and BCT perform equivalently in sycophancy. On jailbreaks, BCT reduced the ASR more than ACT did on an absolute basis, but ACT showed free jailbreak reductions at no significant cost to benign refusals. Further improvements to ACT could make activation supervision more viable for jailbreak reduction.

Initially, we assumed that either it would be too difficult to train consistent activations into the model, or that ACT would be significantly more effective than BCT due to ACT's more mechanistic loss function. This expectation was wrong. In reality, at best, ACT only slightly improved on BCT. At worst, ACT was less effective. ACT *did* tend to have lower side-effects on MMLU and over-refusals.

**Limitations.** Consistency methods assume that the model behaves well on un-augmented data. BCT and ACT widen the distribution of prompts over which the model behaves consistently, so it is possible to train the model to be consistently *unsafe*. In our experiments, we filtered datasets to only prompts where the model's behavior was originally safe to avoid this.

While consistency training is largely unsupervised, it still requires human-curated inputs. The "clean" prompts and augmentation methods creating wrapped prompts still need human guidance. Human designed filters are also needed to filter clean data to prompts which the model behaves well on.

We train models to ignore "irrelevant" information. However, they might mis-generalize by ignoring too much information. Section 4.3.1 finds small impact on instruction-following abilities — one of the main areas we would expect to degrade. However, our evaluations were not exhaustive. Perhaps BCT and ACT degrade attention to detail more severely than suggested by our data.

**Potential benefits from fresh data.** We hypothesized two benefits from consistency training's fresh data. First, consistency training dynamically updates SFT targets to conform to the specification the model is otherwise trained to follow. This benefit comes from the nature of supervised learning.

Second, consistency training avoids capability staleness by training the model on its own response data. We found mixed evidence on the benefits of avoiding capability staleness. Our sycophancy results (Section 4.2.1) provided strong supporting evidence, but our jailbreak results (Section 4.3.1) did not. Why not?

Consider Gemini 2.5 Flash's performance on both tasks. In the sycophancy results, the target data were generated by GPT-3.5-Turbo. This was much older and less capable compared to Gemini 2.0 Flash, the model used for stale data in the jailbreak results. Perhaps 2.0 Flash / Gemma 2 weren't sufficiently outdated to cause degradation on 2.5 Flash / Gemma 3. It may also depend on how capabilities are measured. In jailbreaks, we used over-refusals, but in Appendix E.3 we find SFT (stale) led to lower MMLU accuracy than BCT.

**Future work.** BCT can be viewed as augmenting the training data with "wrapped" (e.g. jailbroken) transformations of existing refusal training points. This could be combined with other data augmentations. For example, our work is compatible with Qi et al. (2024)'s augmentation of beginning a refusal response at a randomly selected token position in a compliant response. More specifically, if the model's wrapped completion is $n$ tokens long, train the model to output its non-wrapped completion starting at a uniformly random position between $t = 0$ and $t = n - 1$. We are also interested in studying the relationship between capability staleness and model degradation. If two models are competitive on benchmarks, can you interchangeably train them to output the other model's clean completions on wrapped prompts without degrading generalization? Is capability staleness largely about benchmark performance, model style, or something else?

## 7 CONCLUSION

We investigated consistency training, a self-supervised framework to make models robust to the irrelevant cues which cause sycophancy and jailbreaks. We compared boosting consistency on model outputs against boosting consistency on internal activations. Bias-augmented Consistency Training defended more strongly against jailbreaks, but Activation Consistency Training had virtually no impact on benign refusals. Although our original intuition on the effectiveness of model internals training was proven incorrect, we find value in viewing problems through the lens of consistency. Consistency methods like BCT simplify training pipelines by removing the independent, bespoke generation of compliant responses. Consistency methods also sidestep the problem of stale data by automatically generating fresh data. We see consistency-across-prompts as a fresh way to address practical alignment problems.

**LLM disclosure** We used LLMs to provide feedback on clarity of the paper, assist with identifying related work, and help generate `matplotlib` plotting code.

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
